# Prenatal screening tests and prevalence of fetal aneuploidies in a tertiary hospital in Thailand

**Preechaya Wongkrajang**[1,2], **Jiraphun Jittikoon**[3], **Sermsiri Sangroongruangsri**[4], **Pattarawalai Talungchit**[5,6], **Pornpimol Ruangvutilert**[5], **Tachjaree Panchalee**[5], **Usa Chaikledkaew**[4,6]*

1 Faculty of Pharmacy, Department of Pharmacy, Social, Economic and Administrative Pharmacy Graduate Program, Mahidol University, Bangkok, Thailand, 2 Faculty of Medicine Siriraj Hospital, Department of Clinical Pathology, Mahidol University Bangkok, Bangkok, Thailand, 3 Faculty of Pharmacy, Department of Biochemistry, Mahidol University, Bangkok, Thailand, 4 Faculty of Pharmacy, Department of Pharmacy, Social Administrative Pharmacy Division, Mahidol University, Bangkok, Thailand, 5 Faculty of Medicine Siriraj Hospital, Department of Obstetrics and Gynecology, Mahidol University, Bangkok, Thailand, 6 Mahidol University Health Technology Assessment Graduate Program, Mahidol University, Bangkok, Thailand

* usa.chi@mahidol.ac.th

**Data Availability Statement:** All relevant data are within the paper.

## Abstract

This study evaluated prenatal screening test performance and the prevalence of common aneuploidies at Siriraj Hospital, Thailand. We collected data from screening tests which are first-trimester test, quadruple test, and noninvasive prenatal tests (NIPT) between January 2016 and December 2020. Thirty percent (7,860/25,736) of pregnancies received prenatal screening tests for aneuploidies disorders, and 17.8% underwent prenatal diagnosis tests without screening. The highest percentage of screening tests was first-trimester test (64.5%). The high-risk results were 4% for first-trimester test, 6.6% for quadruple test, and 1.3% for NIPT. The serum screening tests for trisomy 13 and 18 had no true positives; therefore, we could not calculate sensitivity. For the first-trimester test, the sensitivity for trisomy 21 was 71.4% (95% confidence intervals (CI) 30.3–94.9); specificity for trisomy 13 and 18 was 99.9% (95% CI 99.8–99.9); and for trisomy 21 was 96.1% (95% CI 95.6–96.7). For the quadruple test, the specificity for trisomy 18 was 99.6% (95% CI 98.9–99.8), while the sensitivity and specificity for trisomy 21 were 50% (95% CI 26.7–97.3) and 93.9% (95% CI 92.2–95.3), respectively. NIPT had 100% sensitivity and specificity for trisomy 13, 18 and 21, and there were neither false negatives nor false positives. For pregnant women < 35 years, the prevalence of trisomy 13, 18, and 21 per 1,000 births was 0.28 (95% CI 0.12–0.67), 0.28 (95% CI 0.12–0.67), and 0.89 (95% CI 0.54–1.45), respectively. For pregnant women ≥35 years, the prevalence of trisomy 13, 18, and 21 per 1,000 births was 0.26 (95% CI 0.06–1.03), 2.59 (95% CI 1.67–4.01), and 7.25 (95% CI 5.58–9.41), respectively. For all pregnancies, the prevalence of trisomy 13, 18, and 21 per 1,000 births was 0.27 (95% CI 0.13–0.57), 0.97 (95% CI 0.66–1.44), 2.80 (95% CI 2.22–3.52), respectively.

**Funding:** This study receives funding support from the Health Systems Research Institute (HSRI). The funders had no role in study design, data collection and analysis, decision to publish, or preparation of the manuscript.

**Competing interests:** The authors have declared that no competing interests exist.

## Introduction

Aneuploidies or chromosomal abnormalities are defined as absent or extra chromosomes, including deletions or translocations, and these abnormalities occur in around 0.5%–1.0% live births. In humans, each cell contains 46 chromosomes (23 pairs). During meiosis, gamete cells (sperm and egg cells) are formed 23 single chromosomes per cell. Upon fertilization, the paternal and maternal gamete cells combine, resulting in a diploid cell with 23 pairs of chromosomes (46 chromosomes). Errors in the meiotic segregation process lead to chromosomal abnormalities [1]. Advanced maternal age can increase the risk of aneuploidies [2], and the most common types are trisomy 21, 18, and 13 [3]. These genetic abnormalities can result in many health problems, including intellectual and developmental disorders, cardiovascular diseases, gastrointestinal defects, and other endocrine abnormalities [4,5].

The incidence and prevalence of trisomy 21 (T21), or Down syndrome in European countries are 1/700–800 live births [3] and 2.2/1,000 live births, respectively [6]. In Thailand, the trisomy 21 incidence and prevalence rates per live births are 1/800–1,100 [7] and 1.21/ 1,000, respectively [8]. In European countries, the trisomy 18 (T18) incidence and prevalence rates for live births are 1/3,000–8,000 [4,5] and 0.5/1,000 [6]; for trisomy 13 (T13), or Patau syndrome, the rates per live births are 1/12,000–16,000 [4,5] and 0.2/1,000 live births [6]. However, there have not been any investigations of T18 or T13 in Thailand.

Early detection of fetal chromosomal aneuploidies can assist parents in making pregnancy decisions [1]. Current prenatal screening methods include serum screening, nuchal translucency (NT) using ultrasound, and genetic screening [3]. The first-trimester screening test (FTS), which can be performed at 10–13 weeks gestational age, uses two serum analytes (pregnancy-associated plasma protein A (PAPP-A) and free β-human chorionic gonadotropin (free β hCG)) and NT measurement taken during ultrasound. A previous study has shown the FTS detection rate to be 82%–87% for T21, 97% for T18, and 84% for T13 [9], respectively. The quadruple test (Quad test), which can be performed during the second trimester at 15–22 weeks, measures four serum markers: α-fetoprotein (AFP), free β-human chorionic gonadotropin (free β-hCG), unconjugated estriol (uE3), and inhibin A. The Quad test detects T21 at a rate of 81% [3]. The trisomy risk assessment uses results from these serum tests results and other factors such as age, weight, and race.

In recent years, noninvasive prenatal testing (NIPT), a new test to identify fetal chromosomal abnormalities, has been developed to detect fetal cell-free DNA (cfDNA) in maternal plasma [10], and it can be performed from nine weeks gestational age up to delivery. A meta-analysis of the accuracy of universal NIPT yielded a detection rate of more than 99% for T21, 90% for T18, and 60% for T13 [11].

Since 2019, prenatal screening by serum screening tests in high risk populations (advance maternal age who is pregnant woman age ≥ 35 years) has been included in Thailand's Universal Health Coverage, which covers approximately 80% of the Thai population, and the Thai government plans to extend the prenatal screening test policy to all pregnant women in the near future. However, data on the prevalence of aneuploidies in Thailand have been minimal. There have been only two studies on prenatal screening for T21 in Thailand's southern [7] and northern regions [12]. There have been no studies in Thailand on prenatal screening as well as prevalence for T18 and T13. These data are crucial for the economic evaluation and budget impact analysis before the new Thai prenatal screening policy is implemented.

Hence, we investigated the performance of prenatal screening tests and prevalence of T13, 18 and 21 in the Department of Obstetrics and Gynecology at Siriraj Hospital, Thailand's largest teaching hospital, where approximately 9,000–10,000 pregnant women annually receive antenatal care.

## Materials and methods

### Study population

We obtained the Siriraj Hospital's prenatal screening test data from the records of the Department of Obstetrics and Gynecology and laboratory data from the laboratory information system of the Department of Clinical Pathology for the period January 2016 to December 2020. The medical records were reviewed with the approval of the Siriraj Institutional Review Board (SIRB) (MU-MOU COA 657/2021). The inclusion criteria were Thai ethnicity, singleton pregnancy, and attending an antenatal care clinic before 20 weeks of gestation; the exclusion criteria were incomplete data.

### Assay methods

PAPP-A, free β-hCG, AFP, and uE3 were measured by B·R·A·H·M·S KRYPTOR compact plus (Thermo Fisher Scientific, Hennigsdorf, Germany) using an immunofluorescent assay. Inhibin A was analyzed by Ansh Labs (Medical Center Boulevard, Webster, TX, USA) using an enzyme-linked immunosorbent assay principle. All serum biomarkers were collected and analyzed by the Department of Clinical Pathology, Faculty of Medicine, Siriraj Hospital. The risk of the FTS and the quadruple test was calculated and classified into high and low risk, using the cut off limit of 1:250 based on the Caucasian reference ranges (built-in). For the NIPT results, the data were from various manufacturers such as BGI Genomics Co., Ltd, (China), F. Hoffmann-La Roche Ltd (Switzerland), Bangkok Cytogenetics Center Co., Ltd (Thailand), Faculty of Medicine Siriraj Hospital, Mahidol university (Thailand) and Faculty of Medicine Ramathibodi Hospital, Mahidol university (Thailand).

### Outcome

The outcome of interest was identifying newborns diagnosed with T13, 18 and 21 that had been confirmed prenatally by an invasive prenatal diagnosis test (amniocentesis, chorionic villus sampling, or cordocentesis) or postnatal diagnosis by karyotyping.

### Statistical analysis

Statistical analysis was performed using Microsoft Excel 2019 (Microsoft, Redmond WA, USA). Continuous values were expressed as mean and standard deviation (SD). Categorical data were calculated as frequency and percentage. The prevalence rate was calculated by combining live births, abortions, and pregnancy terminations in the numerator and denominator.

The performance of the FTS, quadruple test, and NIPT was analyzed in terms of sensitivity, specificity, positive predictive value, negative predictive value, accuracy, and prevalence of T13, 18 and 21; 95% confidence intervals (CI) were also calculated.

## Results

Of 46,380 eligible pregnancies, 25,736 met the inclusion criteria. The maternal ages ranged 12–52 years, with a mean of 30.1 and an SD of 5.9 years. Thirty percent (7,726/25,736) of the women were ≥35 years old. Of all cases, the rate of spontaneous abortion was 1,788/25,632 (7%), and the procedure-related loss among women undergoing prenatal diagnostic tests (PND) was 14/5,152 (0.27%).

Fig 1 summarizes the overall design and workflow of participant recruitment. Thirty percent (7,861/25,736) of pregnant women received prenatal screening tests, and 17.8% (4,587/25,736) underwent PND without screening. Among the screening tests, FTS had the highest use rate at 64.5% (5,070/7,861), while the Quad test had the lowest rate at 13% (1,002/7,861). In this study, FTS, Quad test, and NIPT were categorized as high-risk at 4%, 6.6%, and 1.3%,

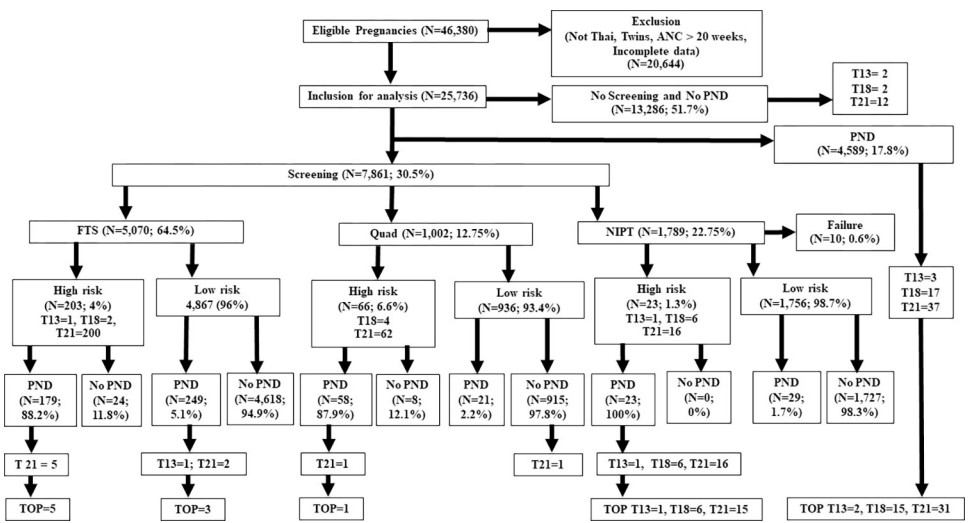

**Fig 1. Flow chart of participants recruitment and overall design.** FTS, first-trimester screening test; Quad, quadruple test; NIPT, non-invasive prenatal testing; PND, prenatal diagnosis test; TOP, Termination of pregnancy.

respectively. PND testing was performed at a rate of 87.9%–100% in the high-risk group but only 1.7%–5.1% in the low risk group due to advanced maternal age, fetal anomaly, or anxiety. Using FTS, the trisomy detection rate was T21 (7 cases) and T13 (1 case). For the Quad test, the detection rate for T21 was two cases. The NIPT method had a failure rate of 0.6% (10/1,789); however, NIPT detected 23 trisomy cases: T13 (1 case), T18 (6 cases), and T21 (16 cases), and 94% (31/33) were terminated. In the PND group, 57 cases with trisomies were detected, and the termination rate was 82.4% (48/57).

Table 1 shows the performance of the prenatal screening tests. There were no true positive values for T13 and T18 using serum biomarkers, therefore we could not calculate sensitivity and positive predictive values (PPV). However, the specificity of T13 detected with FTS was 99.9% (95% CI 99.8–99.9), negative predictive value (NPV) was 99.9% (95% CI 99.8–99.9), and accuracy was 99.9%; for T18, specificity NPV, and accuracy were 99.9% (95% CI 99.8–99.9), 100% (95% CI 99.9–100), and 99.9%, respectively. For FTS, the sensitivity, specificity, PPV, NPV, and accuracy for T21 were 71.4% (95% CI 30.3–94.9), 96.1% (95% CI 95.6–96.7), 2.5% (95% CI 0.92–6.1), 99.9% (95% CI 99.8–99.9), and 96.1%, respectively.

For the Quad test results, the specificity, NPV, and accuracy for T18 were 99.6% (95% CI 98.9–99.8), 100% (95% CI 99.5–100), and 99.6%, respectively; for T21, results were sensitivity (50%, 95% CI 26.7–97.3), specificity (93.9%, 95% CI 92.2–95.3), PPV (1.6%, 95% CI 0.08–9.8), NPV (99.9%, 95% CI 99.3–99.9), and accuracy (93.8%).

The NIPT screening had 100% results for T13, 18 and 21 for sensitivity, specificity, PPV, NPV, and accuracy and there were no false negative or false positive values.

The prevalence rates for T13, 18 and 21 are shown in Table 2. The prevalence T13, 18 and 21 per 1,000 births in pregnant women <35 years was 0.28 (95% CI 0.12–0.67), 0.28 (95% CI 0.12–0.67), and 0.89 (95% CI 0.54–1.45), respectively and termination of pregnancy approximately 60%. For pregnant women ≥35 years, prevalence rates per 1,000 births of T13, 18 and 21 were 0.26 (95% CI 0.06–1.03), 2.59 (95% CI 1.67–4.01), 7.25 (95% CI 5.58–9.41), respectively, and termination of pregnancy ranged from 80% to 100%. For all pregnant women, the prevalence rates per 1,000 births of T13, 18 and 21 were 0.27 (95% CI 0.13–0.57), 0.97 (95% CI 0.66–1.44), 2.80 (95% CI 2.22–3.52), respectively, and termination of pregnancy ranged from 70% to 85% (Table 2).

**Table 1. Performance of prenatal screening test.**

| | N | TP no. | TN no. | FP no. | FN no. | Sensitivity (%) TP/(TP+FN) *100 | 95% CI | Specificity (%) TN/(TN+FP) *100 | 95% CI | PPV(%) TP/(TP +FP)*100 | 95% CI | NPV (%) TN/(TN +FN)*100 | 95% CI | Accuracy (%) TP+TN/(TN+FP +FN+TP) *100 |
|---|---|---|---|---|---|---|---|---|---|---|---|---|---|---|
| First-trimester screening test (total = 5,070) | | | | | | | | | | | | | | |
| Trisomy 13 | 1 | 0 | 5,068 | 1 | 1 | NA | NA | 99.9 | 99.8–99.9 | NA | NA | 99.9 | 99.8–99.9 | 99.9 |
| Trisomy 18 | 0 | 0 | 5,068 | 2 | 0 | NA | NA | 99.9 | 99.8–99.9 | NA | NA | 100 | 99.9–100 | 99.9 |
| Trisomy 21 | 7 | 5 | 4,868 | 195 | 2 | 71.4 | 30.3–94.9 | 96.1 | 95.6–96.7 | 2.5 | 0.92–6.1 | 99.9 | 99.8–99.9 | 96.1 |
| Total | 8 | 5 | 4,864 | 198 | 3 | 62.5 | 25.9–89.8 | 96.1 | 95.5–96.6 | 2.4 | 0.91–5.9 | 99.9 | 99.8–99.9 | 96 |
| Quadruple test (total = 1,002) | | | | | | | | | | | | | | |
| Trisomy 18 | 0 | 0 | 998 | 4 | 0 | NA | NA | 99.6 | 98.9–99.8 | NA | NA | 100 | 99.5–100 | 99.6 |
| Trisomy 21 | 2 | 1 | 939 | 61 | 1 | 50 | 26.7–97.3 | 93.9 | 92.2–95.3 | 1.6 | 0.08–9.8 | 99.9 | 99.3–99.9 | 93.8 |
| Total | 2 | 1 | 935 | 65 | 1 | 50 | 26.7–97.3 | 93.5 | 91.7–94.8 | 1.5 | 0.07–9.3 | 99.9 | 99.3–99.9 | 93.4 |
| NIPT (total = 1,778) | | | | | | | | | | | | | | |
| Trisomy 13 | 1 | 1 | 1,778 | 0 | 0 | 100 | 5.5–100 | 100 | 99.7–100 | 100 | 5.5–100 | 100 | 99.7–100 | 100 |
| Trisomy 18 | 6 | 6 | 1,773 | 0 | 0 | 100 | 51.7–100 | 100 | 99.7–100 | 100 | 51.7–100 | 100 | 99.7–100 | 100 |
| Trisomy 21 | 16 | 16 | 1,763 | 0 | 0 | 100 | 75.9–100 | 100 | 99.7–100 | 100 | 75.9–100 | 100 | 99.7–100 | 100 |
| Total | 23 | 23 | 1,756 | 0 | 0 | 100 | 82.2–100 | 100 | 99.7–100 | 100 | 82.2–100 | 100 | 99.7–100 | 100 |

NA, Not applicable; TP, true positive; FP, false positive; TN, true negative; FN, false negative; PPV, positive predictive value; CI, confidence interval.

## Discussion

This study looked at the pregnant women population of the largest hospital in Thailand (Siriraj Hospital) and showed the performance of prenatal screening tests for T13, 18 and 21 and the prevalence of these syndromes. Our study demonstrated the uptake rate for prenatal screening

**Table 2. Prevalence of trisomy.**

| | <35 years (N = 18,010) | | | | ≥35 years (N = 7,726) | | | | Total (N = 25,736) | | | |
|---|---|---|---|---|---|---|---|---|---|---|---|---|
| Prevalence of trisomy | Total case (N) | Prevalence /1,000 births | 95% CI | TOP (%) | Total case (N) | Prevalence /1,000 births | 95% CI | TOP (%) | Total cases (N) | Prevalence /1,000 births | 95% CI | TOP (%) |
| Trisomy 13 | 5 | 0.28 | 0.12–0.67 | 60 | 2 | 0.26 | 0.06–1.03 | 100 | 7 | 0.27 | 0.13–0.57 | 71 |
| Trisomy 18 | 5 | 0.28 | 0.12–0.67 | 60 | 20 | 2.59 | 1.67–4.01 | 90 | 25 | 0.97 | 0.66–1.44 | 84 |
| Trisomy 21 | 16 | 0.89 | 0.54–1.45 | 69 | 56 | 7.25 | 5.58–9.41 | 77 | 72 | 2.80 | 2.22–3.52 | 75 |
| Total trisomy | 26 | 1.44 | 0.98–2.12 | 65 | 78 | 10.10 | 8.10–12.59 | 81 | 104 | 4.04 | 3.34–4.90 | 77 |

CI, confidence interval; TOP, Termination of pregnancy.

around 30% and prenatal diagnosis without screening of approximately 20%. Prenatal screening tests for fetal chromosomal abnormalities are not compulsory in Thailand for younger pregnant women <35 years old, and they would be offered a choice of either a serum screening test or a NIPT; whereas advanced maternal age pregnancies (≥35 years old) would be given prenatal diagnosis, serum screening or NIPT.

These uptake rates are relatively low compared to other developed countries. In the United States (US), the uptake rate for serum screening tests increased from 22% in 1988 to 72% in 2012 [13]. The uptake rate of screening tests in Europe is around 80%–90% due to the national policy offered to all pregnant women in Belgium, Denmark, Finland, France, and Switzerland [14]. However, the uptake rate in the Netherlands was around 30%–50% due to the additional cost of screening and people considering that Down syndrome is not a severe condition for pregnancy termination [14–16]. In Australia, the screening uptake varied from 44.9% [17] to 83% [18,19]. However, a lower uptake rate for serum screening tests (35%–40%) was found in women <35 years old [20].

In Asian countries like Taiwan, due to the national policies of prenatal diagnosis for pregnant women ≥35 years old and serum screening for younger women <35, the invasive procedure rate in women ≥35 years from 2006 to 2014 was approximately 90%. After introducing NIPT in 2015, the invasive procedure rate for all women increased from 14.7% in 2006 to 25% in 2019. The rate of prenatal diagnosis dropped from 90% to 70% in women ≥35 years old [21]. A study from Israel also demonstrated that serum screening, chorionic villus sampling, amniocentesis procedures decreased following NIPT introduction by 48.7%, 77.2%, and 52.5%, respectively [22]. This is consistent with the US and Hong Kong studies that showed a decline in the number of invasive procedures, around 17% and 26% after NIPT implementation [23,24].

Unfortunately, our study showed no true positives for T13 and T18 from the FTS and Quad tests, thus we could not calculate sensitivity or detection rate. Table 3 shows other countries' detection and false positive rates (FPR). For FTS, the detection rate of T13 ranged 71.9%–84%, FPR ranged 0.5%–5%, T18 ranged 71.9%–97%, and the FPR ranged 0.5%–5%. The detection rate of T21 ranged 71.4%–91.7%, and the FPR ranged 0.5%–7%. The Quad test had a slightly lower detection rate and higher FPR. The detection rate of T21 ranged 50%–76.2%, and FPR ranged 5.1%–9.2% [12,17,25–27].

The different formula used by each laboratory to calculate the risk of fetal aneuploidies could explain the reasons for no true positive result of T13 and T18 and the lower rate of detection of T21. These algorithm models use maternal age, serum of biochemical parameters, and fetal ultrasound examinations. In addition, factors such as gestational age, weight, race, maternal smoking, number of fetuses, and diabetic status can affect the level of the maternal serum biochemical analytes, and these inaccurate data can lead to a wrong estimated risk [9]. Moreover, a certified ultrasonographer must perform NT measurements by ultrasound and participate in ongoing quality control programs [3]. Moreover, we used the Caucasian reference range cut-offs, which might not be appropriate for an Asian population. Wanapirak et al. and Pranpanus et al. demonstrated a better performance for the serum screening test when it was reclassified from the Caucasian reference ranges to Thai reference ranges [12,29].

This study's NIPT performance had excellent results since there were no false positives or false negatives. An earlier meta-analysis about NIPT showed pool sensitivity in unselected pregnant women to be 99%, 90.9%, and 65.1% for T21, T18, and T13, respectively, and the specificity of all trisomy was 99.9% [11]. In Thailand, the sensitivity of T13, 18 and 21 was 100%, and the specificity was 99.9% [30]. The advantage of NIPT is not only highly sensitive and specific results for the common fetal aneuploidies but also fewer false positives.

**Table 3. Detection rate and false positive rate of serum screening tests from other countries.**

| Study | Countries and regions | Year of Study | First-trimester screening test | | | Quadruple test | | |
|---|---|---|---|---|---|---|---|---|
| | | | Cutoff | Detection rate (%) | False positive rate (%) | Cut off | Detection rate (%) | False positive rate (%) |
| Trisomy 13 | | | | | | | | |
| The present study | Siriraj Hospital, Bangkok, Thailand | 2016–2020 | 1:250 | NA | 0.02 | No estimate | No estimate | No estimate |
| Kagan et al [25] | United Kingdom | 2008 | - | 84 | 0.5 | No estimate | No estimate | No estimate |
| Wright et al. [27] | United Kingdom | 2006–2012 | 1:300 | 71.9 | 4.7 | No estimate | No estimate | No estimate |
| Trisomy 18 | | | | | | | | |
| The present study | Siriraj Hospital, Bangkok Thailand | 2016–2020 | 1:250 | NA | 0.04 | 1:250 | NA | 0.4 |
| Kagan et al [25] | United Kingdom | 2008 | - | 97 | 0.5 | No estimate | No estimate | No estimate |
| Wright et al. [27] | United Kingdom | 2006–2012 | 1:300 | 71.9 | 4.7 | No estimate | No estimate | No estimate |
| Trisomy 21 | | | | | | | | |
| The present study | Siriraj Hospital, Bangkok Thailand | 2016–2020 | 1:250 | 71.4 | 3.9 | 1:250 | 50 | 6.1 |
| Kagan et al [25] | United Kingdom | 2008 | | 91 | 5.0 | No estimate | No estimate | No estimate |
| Maxwell et al. [17] | Western Australia | 2005–2006 | 1:300 | 80.9 | 3.4 | 1:300 | 67 | 5.1 |
| Maxwell et al. [26] | Western Australia | 2005–2009 | 1:300 | 82 | 3.2 | No estimate | No estimate | No estimate |
| Wright et al. [27] | United Kingdom | 2006–2012 | 1:300 | 91.7 | 4.7 | No estimate | No estimate | No estimate |
| Wanapiraet al [12] | Northern part of Thailand | 2011–2016 | 1:250 Thai Reference Ranges | 79.2 | 6.8 | 1:250 | 76.2 | 9.2 |
| Kaewsuksai et al [28] | Songkhla, Thailand | 2015–2016 | No estimate | No estimate | No estimate | 1:250 | 75 | 8.6 |

Our study showed a high risk from NIPT at 1.3% with no false positives, which was consistent with other studies (1.3%–2.2%) [30–32]. NIPT results in fewer PND compared to the serum screening test, which showed a high-risk of around 5%. The very high rate of PND due to the false positive rate is not only associated with fetal losses but also the necessity of laboratories for chromosome testing [12]. Our study found NIPT failures to be 0.6%. The test failure rate was about 0.08%–3% [31,33]. The reason may be from procedures, early gestational age (<9–10 weeks), methods, a genetic condition, high body mass index, increased maternal age, race, and other factors [3]. NIPT is a reliable screening test for T13, 18 and 21 but not for structural or other abnormal chromosomal defects. Therefore, NIPT should be performed along with an ultrasound examination [33]. If anomalies are found on ultrasound without evidence of T13, 18 and 21 abnormalities, PND should be offered to detect chromosomal abnormalities beyond the scope of common trisomy disorders [33–35].

Table 4 shows prevalence of trisomy in other countries worldwide.

Our study is the first to demonstrate the prevalence of T13 and T18 abnormalities in Thailand. The prevalence of T13 disorders was 0.27/1,000 births, similar to results from other countries. The prevalence of T13 disorders ranged 0.13–0.37/1,000 live births [6,36–41]. Seventy percent of pregnancies with T13 disorders were terminated, which was comparable to other studies (57.2%–77%) [6,36,40,41].

**Table 4. Prevalence of trisomy in countries worldwide.**

| Study | Countries and regions | Year of Study | Total births | Maternal age ≥35 y (%) | Prenatal screening (%)/ Prenatal diagnosis (%) | Prevalence per 1,000 births | TOP (%) or per 1,000 births |
|---|---|---|---|---|---|---|---|
| Trisomy 13 | | | | | | | |
| The present study | Siriraj Hospital, Bangkok Thailand | 2016–2020 | 25,736 | 30.5 | Prenatal screening: 30% of overall women Prenatal diagnosis: 17.8% of overall women | <35 yr.: 0.28 ≥35 yr.: 0.26 Total: 0.27 | 60% 100% 7% |
| ICBDSR Goel N et al [36] | USA+Europe+ Iran + Israel | 1974–2015 | 16,793,914 | - | - | 0.17 | - |
| BINOCAR Savva et al [37] | UK+Australia | 1989–2004 | 4.5 million | - | - | 1989–1996: 0.12 1997–2004: 0.14 | 57.2% |
| EUROCAT Loane et al [6] | Europe | 1990–2009 | 6,117,757 | 1990–1999: 15.5 2000–2009:18.2 | - | 0.20 | 70.7% |
| MACDP Crider et al [38] | USA | 1994–2003 | | | 70.8% of cases were detected by Prenatal diagnosis but not mention overall % of Prenatal screening | <35 yr.: 0.12 ≥35 yr.:0.36 Total: 0.16 | 60.8% |
| NBDPN Parker et al [39] | USA | 2004–2006 | 4,038,506 | - | - | 0.13 | - |
| NDSCR Springett et al [40] | England and Wales | 2005–2012 | - | - | 90% of cases were detected by Prenatal screening but not mention overall % of Prenatal screening | 0.28 | 77% |
| McDonnell et al. [41] | East of Ireland | 2011–2013 | 80,894 | - | 93% of cases were detected by Prenatal screening but not mention overall % of Prenatal screening | <35 yr.: 0.01–0.03 ≥35 yr.: 0.05–0.15 Total: 0.37 | 70% |
| Trisomy 18 | | | | | | | |
| The present study | Siriraj Hospital, Bangkok Thailand | 2016–2020 | 25,736 | 30.5 | Prenatal screening: 30% of overall women Prenatal diagnosis: 17.8% of overall women | <35 yr.: 0.28 ≥35 yr.: 2.59 Total: 0.97 | 60% 90% 84% |
| ICBDSR Goel N et al [36] | USA+Europe+ Iran + Israel | 1974–2015 | 16,793,914 | - | - | 0.41 | - |
| BINOCAR Savva et al [37] | UK+Australia | 1989–2004 | 4.5 million | - | - | 1989–1996: 0.18 1997–2004: 0.22 | 59.2% |
| EUROCAT Loane et al [6] | Europe | 1990–2009 | 6,117,757 | 1990–1999: 15.5 2000–2009:18.2 | - | 0.5 | 70.5% |
| MACDP Crider et al [38] | USA | 1994–2003 | - | - | 76.1% of cases were detected by Prenatal diagnosis but not mention overall % of Prenatal screening | <35 yr.: 0.23 ≥35 yr.:1.35 Total: 0.40 | 59.7% |
| NBDPN Parker et al [39] | USA | 2004–2006 | 4,038,506 | - | - | 0.27 | - |
| NDSCR Springett et al [40] | England and Wales | 2005–2012 | - | - | 90% of cases were detected by Prenatal screening but not mention overall % of Prenatal screening | 0.70 | 74% |
| McDonnell et al [41] | East of Ireland | 2011–2013 | 80,894 | | 96% of cases were detected by Prenatal screening but not mention overall % of Prenatal screening | <35 yr.: 0.01–0.05 ≥35 yr.: 1.5–5.3 Total: 0.93 | 52% |
| Trisomy 21 | | | | | | | |

(*Continued*)

**Table 4.** (Continued)

| Study | Countries and regions | Year of Study | Total births | Maternal age ≥35 y (%) | Prenatal screening (%)/ Prenatal diagnosis (%) | Prevalence per 1,000 births | TOP (%) or per 1,000 births |
|---|---|---|---|---|---|---|---|
| The present study | Siriraj Hospital, Bangkok Thailand | 2016–2020 | 25,736 | 30.5 | Prenatal screening: 30% of overall women<br>Prenatal diagnosis: 17.8% of overall women | <35 yr.: 0.89<br>≥35 yr.: 7.25<br>Total: 2.80 | 69%<br>77%<br>75% |
| Siripoonya et al [42] | Ramathibodi Hospital, Bangkok, Thailand | 1969–1978 | 46,276 | - | - | 0.89 | - |
| Takeuchi et al. [43] | Japan | 1980–1999 | 108,166 | - | - | 1.52 | - |
| Rudolf et al [44] | Slovenia | 1981–2012 | - | - | 2012: Prenatal screening nearly 80% | 1981: 0.54<br>2012: 2.61 | - |
| BINOCAR Savva et al [37] | UK+Australia | 1989–2004 | 4.5 million | - | - | 1989–1996: 1.53<br>1997–2004:1.94 | - |
| EUROCAT Loane et al. [6] | Europe | 1990–2009 | 6,117,757 | 1990–1999: 15.5<br>2000–2009:18.2 | 70% of cases were detected by Prenatal screening but not mention overall % of Prenatal screening | 2.20 | 46.9% |
| De Graaf et al. [45] | the Netherlands | 1991–2015 | - | - | They mentioned prenatal screening or diagnosis but not stated the percentage | 1991: 1.56<br>2015: 2.26 | 1991: 22%<br>2015: 50% |
| ICBDSR Cocchi G et al. [46] | USA+Canada+Europe +Australia+ Israel | 1993–2004 | 1993: 1,554,529<br>2004: 1,564,501 | 1993: 10.89<br>2004: 18.77 | - | 1993: 1.31<br>2004: 1.82 | 1993: 0.48/ 1,000 births<br>2004: 0.99/ 1,000 births |
| Jou et al. [47] | Taiwan of China | 1993–2001 | 1,331,616 | 1993: 4.8<br>2001: 8.3 | <35 yr.: 65–85%<br>≥35 yr.: 25.3–70.7% | 0.63 | - |
| Acikbas et al [48] | Turkey | 1994–2010 | - | - | - | 0.99 | - |
| Wang et al [49] | China | 2001–2004 | 15,120 | - | Prenatal screening: 100% of overall women | 1.58 | - |
| NBDPN Parker et al [39] | USA | 2004–2006 | 4,038,506 | - | - | 1.45 | - |
| Maxwell et al. [17] | Western Australia | 2005–2006 | 59,999 | 20.3 | Prenatal screening: 56.6% of overall women | 1.62 | - |
| Glivetic et al [50] | Croatia | 2009–2012 | 171,140 | - | - | 0.70 | - |
| Jaruratanasirikul et al [7] | Southern Thailand (Songkhla, Phatthalung, Trang) | 2009–2013 | 186,393 | 2009: 14.7<br>2013:15.5 | 35% of cases were detected by Prenatal diagnosis but not mention overall % of Prenatal screening | <35 yr.: 0.45–0.88<br>≥35 yr.: 4.74<br>Total: 1.21 | 34.1% |
| McDonnell R [41] | East of Ireland | 2011–2013 | 80,894 | - | 47% of cases were detected by Prenatal screening but not mention overall % of Prenatal screening | <35 yr.: 0.08–2.06<br>≥35 yr.: 5.55–20.33<br>Total: 3.57 | 31.1% |
| Wanapirak et al [12,51] | Northern part of Thailand | 2011–2016 | 43,216 | - | Prenatal screening: 100% of overall women | <35 yr.: 1.6<br>≥35 yr.:5.7<br>Total: 1.8 | 79.7% |
| Park et al [52] | Korean | 2007–2015 | 4,140,226 | - | - | 0.5 | - |

ICBDSR, International Clearinghouse for Birth Defects Surveillance and Research; EUROCAT, European Surveillance of Congenital Anomalies; BINOCAR, British Isles Network of Congenital Anomaly Registers; NBDPN, National Birth Defects Prevention Network; MACDP, Metropolitan Atlanta Congenital Defects Program; UK: United Kingdom; USA: The United States of America.

The prevalence of T18 was 0.97/1,000 live births. Our prevalence was slightly higher than that of other countries, ranging 0.40–0.93/1,000 live births [6,36–41]. Eighty-four percent of pregnancies with T18 disorders were terminated, which was higher than in other countries (52%–74%) [6,38,40,41].

The prevalence of T21 in this study was 2.80/1,000 births, comparable to the studies with a high percentage of advanced maternal age. Since T21 is more prevalent than T18 and T13, the studies from other countries showed high variation due to the percentages of pregnant women of advanced maternal age and the uptake rate of prenatal screening or diagnosis ranged from 0.63 to 3.57/1,000 live births [6,7,17,41–51]. Seventy-five percent of pregnancies with T21 abnormalities in our study were terminated, which is similar to the study from Wanapirak et al. [12]. However, the termination rate was higher than the study from Ireland (31.1%) [41], Jaruratanasirikul et al. (34.1%) [7], Croatia (38%) [50], and the European network of population-based registries for the epidemiological surveillance of congenital anomalies (EUROCAT) (46.9%) [6].

Our study had several limitations. This is only single site study and we based our risk estimate on Caucasian rather than Thai reference ranges, which could lead to inaccuracies. In addition, we had a higher percentage of advanced maternal age in our study [53], therefore the prevalence of T13, 18 and 21 may be higher than Thai populations in real situation.

## Conclusion

This current study demonstrates the prevalence and performance of prenatal screening test and the prevalence of common aneuploidies in Thailand. Thirty percent of pregnancies received prenatal screening tests for aneuploidies. The highest percentage of screening tests was first-trimester test. For the first-trimester screening test, the sensitivity for trisomy 21 was 71.4%, specificity for trisomy 13, 18, 21 was 99.9%, 99.9% and 96.1%, respectively. For the quadruple test, the specificity for trisomy 18 was 99.6%, while the sensitivity and specificity for T21 were 50% and 93.9%, respectively. NIPT had 100% sensitivity and specificity for all common trisomy. For pregnant women 35 years, the prevalence of performance per 1,000 births was the lowest compared to other groups. Further studies are needed to explore the prevalence of prenatal screening test performance and the prevalence of common aneuploidies from other sites in Thailand. In addition, the risk estimation should be used Thai reference ranges.

## Acknowledgments

The authors gratefully acknowledge Assoc. Prof. Panutsaya Tientadakul, Head of the Department of Clinical Pathology, Faculty of Medicine Siriraj Hospital, Mahidol University for her support and guidance. and Assoc. Prof. Chayawat Phatihattakorn, Department of Obstetrics & Gynaecology, Faculty of Medicine Siriraj Hospital, Mahidol University for his support.

## Author Contributions

**Conceptualization:** Preechaya Wongkrajang, Jiraphun Jittikoon, Sermsiri Sangroongruangsri, Pattarawalai Talungchit, Usa Chaikledkaew.

**Formal analysis:** Preechaya Wongkrajang, Jiraphun Jittikoon, Sermsiri Sangroongruangsri, Pattarawalai Talungchit, Pornpimol Ruangvutilert, Tachjaree Panchalee, Usa Chaikledkaew.

**Investigation:** Preechaya Wongkrajang.

**Methodology:** Preechaya Wongkrajang, Jiraphun Jittikoon, Sermsiri Sangroongruangsri, Pattarawalai Talungchit, Pornpimol Ruangvutilert, Tachjaree Panchalee, Usa Chaikledkaew.

**Project administration:** Usa Chaikledkaew.

**Supervision:** Usa Chaikledkaew.

**Writing – original draft:** Preechaya Wongkrajang, Jiraphun Jittikoon, Sermsiri Sangroongruangsri, Pattarawalai Talungchit, Pornpimol Ruangvutilert, Tachjaree Panchalee, Usa Chaikledkaew.

**Writing – review & editing:** Preechaya Wongkrajang, Jiraphun Jittikoon, Sermsiri Sangroongruangsri, Pattarawalai Talungchit, Pornpimol Ruangvutilert, Tachjaree Panchalee, Usa Chaikledkaew.

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
