## [Decision Letter · Decision Letter 0]

22 Feb 2023

PONE-D-23-00277Prenatal screening tests and prevalence of fetal aneuploidies in a tertiary hospital in ThailandPLOS ONE

Dear Dr. Chaikledkaew,

Thank you for submitting your manuscript to PLOS ONE. After careful consideration, we feel that it has merit but does not fully meet PLOS ONE’s publication criteria as it currently stands. Therefore, we invite you to submit a revised version of the manuscript that addresses the points raised during the review process.

We look forward to receiving your revised manuscript.

Kind regards,

Nur Aizati Athirah Daud, Ph.D.

Academic Editor

PLOS ONE

Journal Requirements:

Reviewers' comments:

Reviewer's Responses to Questions

**Comments to the Author**

1. Is the manuscript technically sound, and do the data support the conclusions?

Reviewer #1: Yes

Reviewer #2: Yes

2. Has the statistical analysis been performed appropriately and rigorously? 

Reviewer #1: Yes

Reviewer #2: Yes

3. Have the authors made all data underlying the findings in their manuscript fully available?

Reviewer #1: Yes

Reviewer #2: Yes

4. Is the manuscript presented in an intelligible fashion and written in standard English?

Reviewer #1: Yes

Reviewer #2: Yes

5. Review Comments to the Author

Reviewer #1: This is an interesting article highlighting various methods of prenatal testing. The title and abstract are appropriate for the content of the text. The authors however should revise and write more about the definition of aneuploidy and chromosomal abnormality in line 51 under the introduction headline.

The author explained very well the methods, analytical part as well as results. Comparison has been made with other studies even though from different populations. The results of their study might contribute towards prenatal advancement mainly for genetic testing and counselling, and as a reference for obstetricians and gynaecologists in Asian countries mainly.

Reviewer #2: The authors present a strong cross sectional sampling from a single hospital on fetal aneuploidies for a Thai population. It is solidly presented, and my comments are relatively minor and more suggestions for stylistic approaches for presentation if I may.

1. Abstract: The abstract presents a lot of paper, which is admirable, but it might be an idea to add an opening sentence or two on why this screening is undertaken for non-expert readers. This is optional, but can help the wide audience of PLOS one better appreciate what work you undertook

2. It would be helpful for authors to define their sensitivity analysis in the methods section is a little more depth. From table 1, I presume they simply counted the fraction of true positives over (true positive plus false negatives) from the data, and this is absolutely fine and correct. However it would be useful for non-experts to have these clearly defined (for an example from another field, table 1 in this paper does so to reduce any ambiguity https://jamanetwork.com/journals/jamanetworkopen/fullarticle/2781509 )

3. With regards to table 1, it might be worth putting total numbers of real cases per item of interest in the table, possible in brackets after the condition as this makes it easier for readers to pick out the denominator. For example, in the first-trimester screening case for T21, there were 5TPs and 2FN, which gives N = 7 cases, which of course is whence sensitivity is derived. But putting total cases somewhere would help, maybe something like: Trisomy 21 (n =7) in the first column etc...

4. In table 3 presenting the results of others, the blacked our entries are a little confusing: am I correct in assuming these are entries for which other studies did not produce data? I think so from the text, but there might a nicer way to present this than just blacking out: text like "No estimate" or similar perhaps?

5. Finally, have you thought about a graphical representation of these results? I think you could plot all your estimates and CIs in a single plot if you wished.

Congratulations again on an interesting paper.

6. PLOS authors have the option to publish the peer review history of their article (what does this mean?). If published, this will include your full peer review and any attached files.

Reviewer #1: No

Reviewer #2: No

---

## [Author Response · Author response to Decision Letter 0]

24 Mar 2023

23 March 2023

Dear Editors,

On behalf of my co-authors, we would like to resubmit the revised manuscript of our study entitled “Prenatal screening tests and prevalence of fetal aneuploidies in a tertiary hospital in Thailand” (Manuscript ID number: PONE-D-23-00277) for your kind consideration to be published on PLOS ONE.

We would like to thank all editors and reviewers for their helpful comments and suggestions. We feel that the revised paper is much further improved as a consequence of their inputs. The next page is a point-by point form explaining how we have responded to each comment raised by the editors and reviewers. 

Should you have any question, please kindly contact me at usa.chi@mahidol.ac.th. Thank you very much for your kind consideration on this manuscript.

Sincerely yours,

Assoc. Prof. Usa Chaikledkaew, Ph.D.

Corresponding author 

Social Administrative Pharmacy Division, Department of Pharmacy and Mahidol University Health Technology Assessment (MUHTA) Graduate Program, Mahidol University

447 Sri-Ayuthaya Road, Rajathevi, Bangkok 10400, Thailand

Tel: 662-644-8679 ext 5317; Fax: 662-644-8694

Email: usa.chi@mahidol.ac.th

Response to Editors and Reviewer:

Journal Requirements:

Response: Thank you very much. We have already checked that our manuscript meets PLOS ONE’s style requirements. 

Response: Thank you very much. We have already reviewed the reference list to ensure that our reference list is complete and correct. 

Reviewer #1: 

This is an interesting article highlighting various methods of prenatal testing. The title and abstract are appropriate for the content of the text. The authors however should revise and write more about the definition of aneuploidy and chromosomal abnormality in line 51 under the introduction headline. The author explained very well the methods, analytical part as well as results. Comparison has been made with other studies even though from different populations. The results of their study might contribute towards prenatal advancement mainly for genetic testing and counselling, and as a reference for obstetricians and gynecologists in Asian countries mainly.

Response: Thank you very much for your kind suggestion. We have added more sentences about definition of aneuploidy and chromosomal abnormality in the introduction section (Line 53-57, Page 3).

“In humans, each cell contains 46 chromosomes (23 pairs) and during meiosis, gamete cells (sperm and egg cells) are formed 23 single chromosomes per cell. Upon fertilization, the paternal and maternal gamete cells combine, resulting in a diploid cell with 23 pairs of chromosomes (46 chromosomes). Errors in the meiotic segregation process can lead to chromosomal abnormalities [1].”

Reviewer #2: 

The authors present a strong cross-sectional sampling from a single hospital on fetal aneuploidies for a Thai population. It is solidly presented, and my comments are relatively minor and more suggestions for stylistic approaches for presentation if I may.

1. Introduction: The Introduction presents a lot of paper, which is admirable, but it might be an idea to add an opening sentence or two on why this screening is undertaken for non-expert readers. This is optional, but can help the wide audience of PLOS one better appreciates what work you undertook

Response: Thank you very much for your kind suggestion. We have added an opening sentence on why this screening is undertaken in the introduction section (Line 68-69, Page 3).

“Early detection of fetal chromosomal aneuploidies can assist parents in making pregnancy decisions [1].”

2. It would be helpful for authors to define their sensitivity analysis in the methods section is a little more depth. From table 1, I presume they simply counted the fraction of true positives over (true positive plus false negatives) from the data, and this is absolutely fine and correct. However it would be useful for non-experts to have these clearly defined (for an example from another field, table 1 in this paper does so to reduce any ambiguity https://jamanetwork.com/journals/jamanetworkopen/fullarticle/2781509 )

Response: Thank you very much for your kind suggestion. We have already added formula of sensitivity, specificity, positive predictive value, negative predictive value, accuracy and total number of trisomy in Table 1 (Page 8).

3. With regards to table 1, it might be worth putting total numbers of real cases per item of interest in the table, possible in brackets after the condition as this makes it easier for readers to pick out the denominator. For example, in the first-trimester screening case for T21, there were 5TPs and 2FN, which gives N = 7 cases, which of course is whence sensitivity is derived. But putting total cases somewhere would help, maybe something like: Trisomy 21 (n =7) in the first column etc...

Response: Thank you very much for your kind suggestion. We have already added total numbers of total cases in the first column of Table 1 (Page 8).

4. In table 3 presenting the results of others, the blacked our entries are a little confusing: am I correct in assuming these are entries for which other studies did not produce data? I think so from the text, but there might a nicer way to present this than just blacking out: text like "No estimate" or similar perhaps?

Response: Thank you very much for your kind suggestion. We have used "No estimate" instead of blacking out in Table 3 (Page 13).

5. Finally, have you thought about a graphical representation of these results? I think you could plot all your estimates and CIs in a single plot if you wished.

Congratulations again on an interesting paper.

Response: Thank you very much for your kind suggestion. However, we would like to present the results in a table format which can show all data in exact details, as these data may be more useful and helpful for obstetricians and gynecologists especially in Asian countries to use as the references.

---

## [Editor Report · Decision Letter 1]

10 Apr 2023

Prenatal screening tests and prevalence of fetal aneuploidies in a tertiary hospital in Thailand

PONE-D-23-00277R1

Dear Dr. Chaikledkaew,

We’re pleased to inform you that your manuscript has been judged scientifically suitable for publication and will be formally accepted for publication once it meets all outstanding technical requirements.

Kind regards,

Nur Aizati Athirah Daud, Ph.D.

Academic Editor

PLOS ONE

---

## [Editor Report · Acceptance letter]

13 Apr 2023

PONE-D-23-00277R1 

Prenatal screening tests and prevalence of fetal aneuploidies in a tertiary hospital in Thailand 

Dear Dr. Chaikledkaew:

I'm pleased to inform you that your manuscript has been deemed suitable for publication in PLOS ONE. Congratulations! Your manuscript is now with our production department. 

Kind regards, 

on behalf of

Dr. Nur Aizati Athirah Daud 

Academic Editor

PLOS ONE